# *Marchantia polymorpha* as a Source of Biologically Active Compounds

**DOI:** 10.3390/molecules30030558

**Published:** 2025-01-26

**Authors:** Filip Nowaczyński, Rosario Nicoletti, Beata Zimowska, Agnieszka Ludwiczuk

**Affiliations:** 1Department of Pharmacognosy with the Medicinal Plant Garden, Medical University of Lublin, 20-093 Lublin, Poland; agnieszka.ludwiczuk@umlub.pl; 2Council for Agricultural Research and Economics, Research Center for Olive, Fruit and Citrus Crops, 81100 Caserta, Italy; 3Department of Plant Protection, University of Life Sciences, 20-069 Lublin, Poland; beata.zimowska@up.lublin.pl

**Keywords:** common liverwort, terpenoids, bisbibenzyls, biological activity, Marchantiaceae

## Abstract

*Marchantia polymorpha* L., also known as common liverwort or umbrella liverwort, is a spore-forming plant belonging to the Marchantiaceae family. This thallose liverwort has gained importance as a model plant, mainly because of its global distribution and easy and rapid in vitro culturing. A review of the literature shows that the major compounds in this species are undoubtedly sesquiterpenoids and bisbibenzyls. Among the sesquiterpenoids, it is worth mentioning cuparenes, chamigranes, and thujopsanes. Compounds belonging to these classes were found in specimens from Japan, China, Poland, Germany, and India and could be the chemical markers of this liverwort species. The key secondary metabolite of *M. polymorpha* is a macrocyclic bisbibenzyl, marchantin A. Marchantin-type aromatic compounds, together with other bisbibenzyls, such as riccardin D, isoriccardin C, or perrottetin E, demonstrated antifungal and antibacterial properties in various studies. In this review, we summarize the current knowledge on the diversity of compounds produced by *M. polymorpha*, emphasizing chemical variability depending on the origin of the plant material. Moreover, the biological activity of extracts obtained from this liverwort species, as well as single secondary metabolites, are described.

## 1. Introduction

Bryophytes are terrestrial, spore-bearing plants that comprise three phyla: liverworts (Marchantiophyta), mosses (Bryophyta), and hornworts (Anthocerophyta). These small nonvascular plants, phylogenetically placed between algae and ferns, are considered the first inhabitants of terrestrial habitats [1]. As the first land plants, they had to cope with adverse environmental conditions; hence, their ability to synthesize many different specialized secondary metabolites is extremely high. Indeed, such ‘chemical weapons’ are necessary for these small plants, since they the lack mechanical protection of higher vascular plants [2]. Among the bryophytes, the chemical constituents of the Marchantiophyta and their biological activity have been studied in great detail. Over the last 40 years, more than 3000 compounds have been found in this group of plants. Many of these products are characterized by unprecedented structures, and some, including the pinguisane-type sesquiterpenoids and the sacculatane-type diterpenoids, have not been found in any other plants, fungi, or marine organisms. This unique chemical composition increases the number of potential applications in medicine and beyond. In fact, the available literature data indicate that liverwort secondary metabolites show antibacterial, antifungal, cytotoxic, insect repellant, enzyme inhibitory, and proapoptotic activities [3,4,5].

*Marchantia polymorpha* L., also known as common liverwort or umbrella liverwort and belonging to the Marchantiaceae family (Figure 1), is the most widely distributed liverwort in the world. It is a cosmopolitan species that occurs from tropical to arctic regions [6,7].

This liverwort species has become one of the most important models for plant biology research and evolutionary genomics due to its relatively simple genome, global distribution, easy in vitro culturing, and unique phylogenetic position as a member of the early land plants [8,9]. As an evolutionary model, *M. polymorpha* contributes to our understanding of the evolution of plant defensive responses and the associated hormonal signaling pathways [9]. At this point, the following questions arise about the biologically active compounds present in this model plant: What do we know about them? Does *Marchantia* have its own characteristic metabolites? And are they used in medicine, horticulture, or for other purposes?

The aim of this paper is to review the available scientific literature concerning both the chemical composition and the biological properties of the most well-known liverwort species, *M. polymorpha*. Special attention was paid to the variability of the chemical composition depending on the origin of the plant material.

## 2. Chemical Diversity of *M. polymorpha*

Liverworts (Marchantiophyta) are plants that produce a wide array of biologically active secondary metabolites. These compounds are accumulated in the oil bodies, which are prominent and highly distinctive organelles uniquely found in liverworts [10]. Oil bodies are present in 95% of all liverwort species and are intracellular organelles bounded by a single unit membrane originating from dilated endoplasmic reticulum cisternae, containing lipophilic globules [11]. In the thallose liverworts like *M. polymorpha*, oil bodies are confined to scattered idioblastic oil body cells, while oil bodies of leafy liverworts are generally present in all cells [12]. The number, size, and colour of oil bodies are species specific. Oil bodies are estimated to serve a protective role for the plant, with their contents postulated to protect the plant against various biotic and abiotic stressors [13].

A review of the literature on the chemical composition of the umbrella liverwort shows that it is characterized by great diversity. The following groups of chemical compounds have been identified so far in *Marchantia*: monoterpenoids, sesquiterpenoids, and diterpenoids; sterols and triterpenoids; and bibenzyls, bisbibenzyls, phenanthrene derivatives, flavonoids, lipids, and other compounds (Table 1).

Such a great chemical diversity of *M. polymorpha*, shown in Table 1, suggests the heterogeneity of this liverwort species. *Marchantia* is characterized by high morphological variability, which has led to the description of many synonymous “species” and subspecific taxa [8]. Based on the analysis of morphological characters, isozyme patterns, RFLP (nrDNA), RAPD markers, and ecological preferences, three subspecies of *M. polymorpha* are distinguished. These are *M. polymorpha* subsp. *polymorpha*, *M. polymorpha* subsp. *montivagans*, and *M. polymorpha* subsp. *ruderalis*. The available literature data indicate that the subsp. *polymorpha* corresponds to the plant previously known as *Marchantia aquatica*, while the subsp. *montivagans* corresponds to the liverwort known as *Marchantia alpestris*. Only the subsp. *ruderalis* corresponds to *M. polymorpha* (sensu stricto). These three taxa differ in habitat, although they sometimes occur sympatrically [55,56,57]. More detailed analyses conducted by Linde and his associates [58] revealed a more complex pattern, with evidence suggesting hybridization and introgression between subspecies.

When reviewing the literature on chemical composition, unfortunately, there is almost no data on the occurrence of plant metabolites in individual subspecies. Most of the publications refer simply to *M. polymorpha* without indicating the subspecies. The data included in Table 1 suggest that most of the metabolites were identified and/or isolated from subsp. *ruderalis*. This is also supported by the ecological preferences of individual subspecies. Boisselier-Dubayle and coworkers [55] showed that *M. polymorpha* subsp. *polymorpha* occurs chiefly in flooded habitats, and subsp. *montivagans* grows in wet habitats in the mountains, while subsp. *ruderalis* colonizes preferentially man-made habitats, implying that, when collecting plant material, it is easiest to encounter *M. ruderalis*.

The phytochemistry of *M. polymorpha* also varies depending on its place of origin. The major chemical compounds contributing to the phytochemical complexity of this liverwort species are distributed among two groups: sesquiterpenoids and bisbibenzyls. A comparison of the chemical composition of the available data concerning subspecies, as well as geographical distribution in relation to the most characteristic metabolites of *M. polymorpha*, are presented in Table 2.

### 2.1. Sesquiterpenoids

*Marchantia polymorpha* is a rich source of terpenoids, in particular those belonging to the sesquiterpene group. Forty-eight sesquiterpenoids belonging to twenty different classes are included in Table 1. The structures of selected sesquiterpenoids characteristic of *M. polymorpha* are presented in Figure 2.

The first sesquiterpenoid reported from *M. polymorpha* was *(S)*-2-hydroxycuparene (=2-cuparenol). Its isolation was conducted in 1974 by Hopkins and Perold [26] from a South African specimen. Two cuparane-type alcohols, cyclopropanecuparenol and its epimer, are the major volatile components of this species. Besides the mentioned alcohols, other cuparanes are present in *M. polymorpha*, namely cuparene and α-, β-, γ-, and δ-cuprenene. Thujopsanes and chamigranes are other sesquiterpenoids characteristic of *M. polymorpha*. They are represented by thujopsene, thujopsan-7β-ol, thujopsenone, α- and β-chamigrene, as well as *ent*-9-oxo-α-chamigrene [12,18]. The data included in Table 2 show that these sesquiterpenoids (cuparanes, chamigranes, and thujopsanes) could be chemical markers of *M. polymorpha* subsp. *ruderalis*. Occasionally, this subspecies can produce metabolites characteristic of a single specimen. In the Polish collection of *M. polymorpha*, acorane-type sesquiterpenoids were identified. The presence of α-neocallitropsene, acorenone B, β-alaskene, and β-acoradiene were confirmed [59,60].

Our recent preliminary data concerning volatile components present in the Serbian *M. polymorpha* subspecies showed that the subspecies *polymorpha* and *montivagans* are very different with reference to sesquiterpene composition. In the case of these subspecies, the presence of cuparane-, chamigrane-, and thujopsane-type compounds was not demonstrated, while compounds belonging to aromadendranes, guaianes, and eudesmanes were identified [17]. A very similar chemical composition was also observed in the sample from Turkey [16]. Although the authors did not specify the subspecies of the specimen studied, it can be inferred that it is not *M. ruderalis*.

### 2.2. Bibenzyls and Bisbibenzyls

Bibenzyls are organic compounds with a C6-C2-C6 skeleton, which are synthesized by the phenylpropanoid pathway, like polyphenols [61]. Common liverwort is reported to produce only a few compounds; among them, it is worth mentioning lunularin and lunularic acid [23,26,62]. Both metabolites are direct precursors in the biosynthesis of marchantin C, a bisbibenzyl, which is later transformed to form marchantin A [63].

Bisbibenzyls are macrocyclic compounds consisting of two bibenzyl units. Acyclic bisbibenzyl compounds are linked once, while the cyclic ones are linked twice. The most important bisbibenzyl found in *M. polymorpha* is marchantin A. It is derived from lunularic acid, with two ether linkages between C_1_–C_2′_ and between C_14_–C_11′_ (Figure 3). The majority of common liverwort specimens contain marchantin A in large amounts. In fact, it was reported to be present in common liverwort from various countries (Table 1). This, however, is not true for South African *M. polymorpha*, which, according to some studies, does not contain marchantin A at all [33]. Its place as the major cyclic bisbibenzyl is taken by marchantin H. Moreover, marchantin E has been isolated form Indian and French specimens [5]. Marchantin A is also commonly found in many other plants from the Marchantiales [64,65] and other Marchantiophyta [66].

Riccardins are another group of cyclic bisbibenzyl compounds present in *M. polymorpha*. Japanese and Indian specimens of *M. polymorpha* contain riccardin C [13,22,40]. Riccardin H, isoriccardin D, and 13,13′-*O*-isopropylidenericcardin D were found in *M. polymorpha* from China [62]. Isoriccardin C was found in Chinese, Indian, and Vietnamese plant material [13,62,67].

Other bisbibenzyls that can be found in common liverwort are perrottetin E and polymorphatin A. Perrottetin E is an acyclic bisbibenzyl found in Indian and Chinese specimens of common liverwort [13,62]. It can be used as a precursor for the synthesis of marchantin- and riccardin-type compounds [33]. Polymorphatin A is a cyclic compound linked with one ether C_1_–C_2′_ linkage and one biphenyl C_12_–C_12′_ linkage. This bisbibenzyl was first found in Chinese *M. polymorpha* [62]. Representatives of riccardins and other bisbibenzyls are presented in Figure 4.

Based on the data presented in Table 2, specimens from European countries and Japan are characterized by the occurrence of only marchantin A derivatives. On the other hand, those from India, China, and Vietnam, as well as South Africa, in addition to marchantin-type compounds, also produce those belonging to the riccardin and perrottetin types.

### 2.3. Other Compounds

Flavonoids are ubiquitous minor components in the Marchantiophyta, including *M. polymorpha* [3,4,68]. The main flavonoid types present in this species are flavone O-glucuronides. Luteolin, apigenin, and their derivatives are the most abundant, as shown in Table 1.

Another interesting biochemical feature of *M. polymorpha* is represented by two polyunsaturated fatty acids, arachidonic acid (ARA, 20:4n-6) and eicosapentenoic acid (EPA, 20:5n-3). Shinmen et al. [54] have reported that culture of M. *polymorpha* contained high amounts of ARA and EPA (92 and 48 mg L^−1^, respectively) under photomixotrophic conditions.

Among other products of *M. polymorpha*, it is worth mentioning monoterpenoids and diterpenoids, sterols and triterpenoids, phenanthrenes, phthalides, and other aromatic compounds. Characteristic diterpenoids can be found in Vietnamese specimens, such as marchanol, belonging to the clerodane-type compounds, and vitexilactone from the labdane group [31].

*Marchantia polymorpha* does not synthesize monoterpenes, apart from limonene, which was reported at the initial stage of growth in cell culture [14].

The sterols and triterpenoids found in common liverwort are similar to those found in the higher plants. Among the sterols, the presence of sitosterol and stigmasterol was confirmed, while among triterpenoids, the occurrence of ursane- and oleanane-type compounds was reported [22,31,33]. Phenanthrene derivatives were found in the field collection of *M. polymorpha* from India [23], as well as from cell cultures in Germany [37]. Finally, the presence of two phthalides, 3R-(3,4-dimethoxybenzyl)-5,7-dimethoxyphthalide and marchatoside, was confirmed in a Vietnamese collection [31].

The chemical structures of selected diterpenoids and triterpenoids, flavonoids, phenanthrenes, and phthalides are presented in Figure 5.

## 3. Biological Activities

Common liverwort has a long history in ethnomedicine [64]. It was used as an antipyretic, antihepatic, antidotal, and a diuretic medicinal plant [69].

Extracts of *M. polymorpha* were repeatedly proven to possess antifungal properties [42,70,71,72,73]. Many fungi are susceptible to growth inhibition when subjected to these extracts, including *Candida albicans* [35,70,74], *Cryptococcus neoformans*, *Tilletia indica*, *Fusarium oxysporum* f.sp. *lini*, *Sclerotium rolfsii*, *Rhizoctonia solani* [42,71], *Alternaria solani* [74], *Fusarium solani* [73], and *Trichophyton mentagrophytes* [35,74]. Studies of activity against *C. albicans* determined that neomarchantin A, riccardin D, and 13,13′-*O*-isopropylidene-riccardin D are the most effective compounds, while marchantin A, B, and E and riccardin H, even if possessing some antifungal activity, were not as effective [70]. The activity also varies depending on the solvent used for extraction. Riccardin D (called by the authors plagiochin E), found in Chinese *M. polymorpha*, exhibits inhibitory properties against *C. albicans*, which were increased when combined with fluconazole. When examined in more detail, this compound was found to reverse the fungal resistance to the azole drug by inhibiting its efflux from *C. albicans* [75]. Transmission electron microscopy showed serious damage in the structure of the yeast cell wall after treatment with plagiochin E. Inhibition of chitin synthesis was detected, deriving from downregulation of the expression of *CHS1* and upregulation of the expression of *CHS2* and *CHS3* [76]. Moreover, exposure to plagiochin E resulted in an elevation of the membrane potential and a decrease of the ATP level in mitochondria, which caused ROS accumulation [77]. This effect induced typical markers of apoptosis in the yeast cells, such as chromatin condensation, nuclear fragmentation, and G_2_/M cell cycle arrest. The latter event was related to downregulation of cyclins (CDC28, CLB2 and CLB4), as well as metacaspase activation [78].

Antibacterial activity of extracts from *M. polymorpha* is also an important subject of studies on common liverwort, although no information has been collected so far with reference to the mechanisms of action. Besides the crude extracts [71,74], marchantin A also exhibits such properties; in fact, its inhibitory effect has been documented on both Gram-positive and Gram-negative bacteria, such as *Acinetobacter calcoaceticus*, *Bacillus cereus*, *Bacillus megaterium*, *Bacillus subtilis*, *Escherichia coli*, *Haemophilus influenzae*, *Listeria monocytogenes*, *Neisseria meningitidis*, *Pasteurella multocida*, *Pseudomonas aeruginosa*, *Proteus mirabilis*, *Staphylococcus aureus*, *Staphylococcus epidermidis*, *Streptococcus pyogenes*, and *Streptococcus viridans* [23,42,62,68,79,80]. Antibacterial effects have been also documented in the case of isoriccardin C [62]. Lines of *M. polymorpha* were also subjected to genetic engineering in order to obtain a mutant with a higher potential for the synthesis of antibacterial compounds [35].

In vitro studies have shown that organic extracts of *M. polymorpha* exhibit cytotoxic activity [81,82], deriving from their content in bioactive products. A more detailed study conducted in 2008 showed that marchantin A induces growth inhibition on the breast cancer cell lines A256, MCF-7, and T47D, based on an antimicrotubular effect which was increased when marchantin A and an Aurora-A kinase inhibitor were used simultaneously [83]. Marchantin A also demonstrated cytotoxicity against the malignant melanoma cell line A375 while having less cytotoxic activity against keratocytes and not affecting tyrosinase activity in a model assay [84]. Lunularin also exhibited potent cytotoxic activity against MCF-7 [62].

Marchantin A exhibits DNA polymerase β-inhibitory and anti-HIV activities [85]. Moreover, along with marchantins B and E, plagiochin A, and perrottetin F, it possesses anti-influenza activity deriving from its targeting of the PA subunit of endonuclease. These products have a 3,4-dihydroxyphenethyl group in common, which is indicative of the importance of this moiety for this kind of bioactivity [86]. Marchantin A was also found to inhibit the proliferation of the erythrocytic stages of two *Plasmodium falciparum* strains, as well as other protozoans, such as *Trypanosoma brucei rhodesiense*, *T. cruzi*, and *Leishmania donovani* [87]. Its antitrypanosomal activity was also documented by another research group, along with marchantin E [88]. However, marchantin A has a low sensitivity index towards the aforementioned parasites, so the therapeutic window is rather narrow [1].

While tested for its antioxidant properties, marchantin A showed free radical scavenging ability [35] depending on its concentration. It also ties in to the anti-inflammatory properties of *M. polymorpha*, originating in its ethnomedicinal uses. Marchantins A, B, D, and E, isoriccardin C, and perrotetin D demonstrated an inhibitory effect on 5-lipoxygenase and cyclooxygenase, key enzymes in the arachidonic acid cascade [89]. The strength of this effect is structure dependent, as marchantin D exhibited lower inhibition toward 5-lipoxygenase. Along with lunularin, isoriccardin C has also displayed strong DPPH radical scavenging activity [62].

A chloroform extract of *M. polymorpha* was postulated to have hepatoprotective properties [90]. When mice were administered with paracetamol in liver-damaging quantities along with marchantin A, the amount of markers of liver damage in mice blood (aspartate transaminase and alanine transaminase) was significantly lower than in the control group administered with paracetamol only, and on par with the group in which paracetamol was administered along with silymarin. Another study showed that flavonoids of *M. polymorpha* can protect liver cells from injuries caused by the administration of carbon tetrachloride [91]. As both compounds induce damage to liver cells with their oxidizing potential, the hepatoprotective effect was postulated to be due to the antioxidant properties of *M. polymorpha* extracts.

Marchantin A, riccardin A, marchantin B, and other compounds from *M. polymorpha* also have an inhibitory effect on lipopolysaccharide production induced by nitric oxide [92]. As nitric oxide is postulated to play a role in the etiology of chronic neurodegenerative diseases [93], this property should be more closely investigated in the future.

Structural similarity between cyclic bisbibenzyl compounds and bisbenzylisoquinoline alkaloids, such as tubocurarine, has led to the investigation of the muscle relaxation properties of marchantin-type compounds [94]. In a study published in 1995 [95], marchantin A was used in comparison to cepharanthine, a muscle relaxant. Both compounds expressed similar properties and were bound to a common receptor, which points to the muscle-relaxing properties of marchantin A likely being owed to the binding of calcium molecules. This may also tie in with the inhibition by marchantin A of calmodulin [95], a protein with activity related to calcium levels in the cell.

## 4. Conclusions and Future Perspectives

Although bryophytes are among the oldest land plants, their usefulness is relatively unknown to most people. There is very little knowledge available about the medicinal properties of bryophytes. An ancient method of determining the medicinal properties of plants was based on the concept of Paracelsus, dealing with the resemblance of plant body parts to the shape and structure of organs in the human or animal body for which it is remedial. As per the abovementioned philosophy, *M. polymorpha* was used to cure hepatic disorders [96]. This liverwort was also used as a medicine for boils and abscesses, perhaps because the young archegoniophore resembles a boil when it first emerges from the thallus [97].

From this perspective *M. polymorpha* became a very interesting case study. This liverwort is present in almost all environments and has a very versatile phytochemical profile, especially including bisbibenzyl compounds. These compounds are almost unique to liverworts, as their presence has so far only been confirmed for plants of the *Primula* genus [1,5]; they have disclosed particular interest in biological activity studies and may be the foundation of new plant medicines or plant protection products.

However, until now, no products derived from *M. polymorpha* are available on the market. Also, to the best of our knowledge, no clinical, pre-clinical or toxicological studies have been carried out so far. Although the exact mode of action of some of the described bioactive compounds remains unknown, *M. polymorpha* and its metabolites could serve as an attractive candidate for therapeutic properties. Further work on the isolation, characterization, structural elucidation, pharmacological evaluation, determination of mode of action, and clinical trial of these active principles could open exciting perspectives in future drug development programs.

The secondary metabolites of *M. polymorpha* endophytes show applicative potential as well. They exhibit selective cytotoxicity toward several cancer cell lines and antiviral properties [15,98], calling for more accurate investigations on their occurrence and bioactivities. As of late, our team is trying to establish the optimal ways to cultivate these bryendophytes and to extract and evaluate their products.

As we managed to summarize in this article, the therapeutic potential of *M. polymorpha* is yet to be fully explored, but, even now, it holds potential for many future studies, which may result in crucial findings.

## Figures and Tables

**Figure 1 molecules-30-00558-f001:**
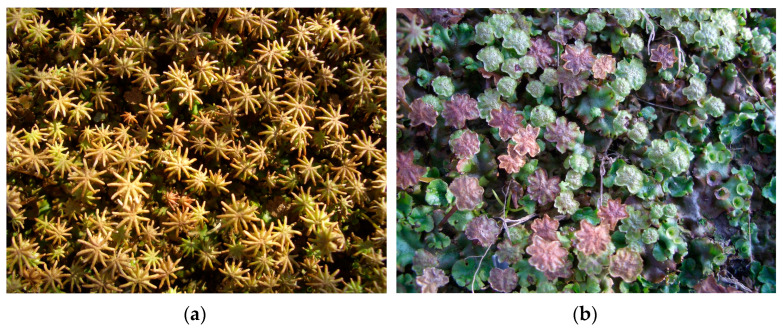
*Marchantia polymorpha*—umbrella liverwort: (**a**) female and (**b**) male plant. (Photos by Prof. Yoshinori Asakawa, Tokushima Bunri University, Japan).

**Figure 2 molecules-30-00558-f002:**
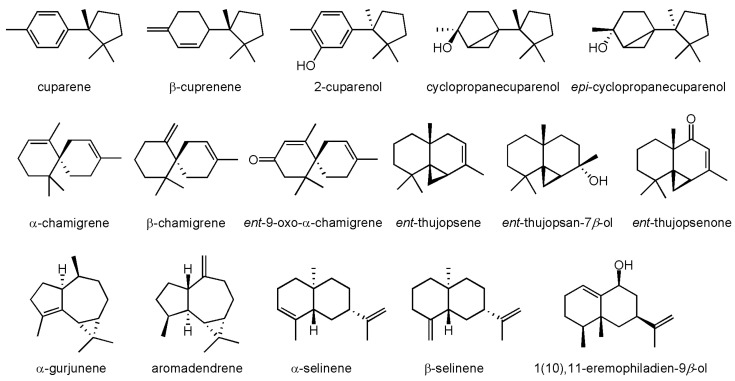
Selected sesquiterpenoids characteristic of *M. polymorpha*.

**Figure 3 molecules-30-00558-f003:**
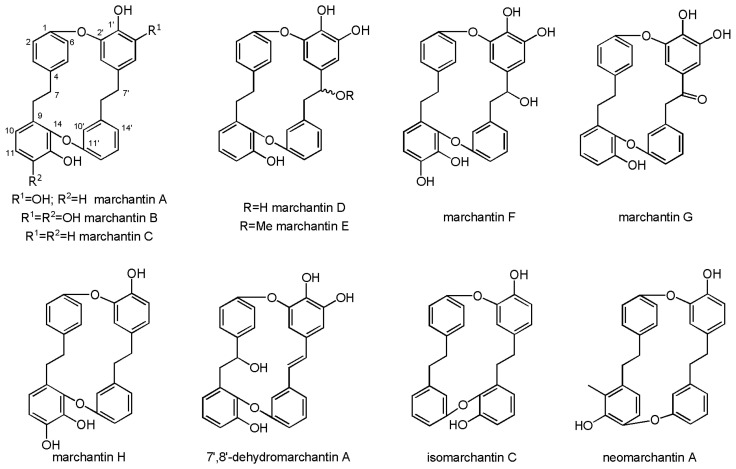
Chemical structures of marchantin-type bisbibenzyls.

**Figure 4 molecules-30-00558-f004:**
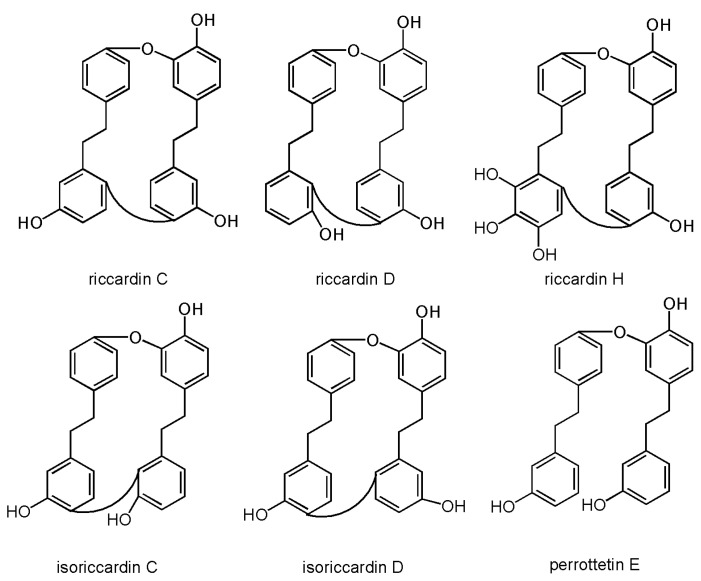
Chemical structures of some riccardins and perrottetin E.

**Figure 5 molecules-30-00558-f005:**
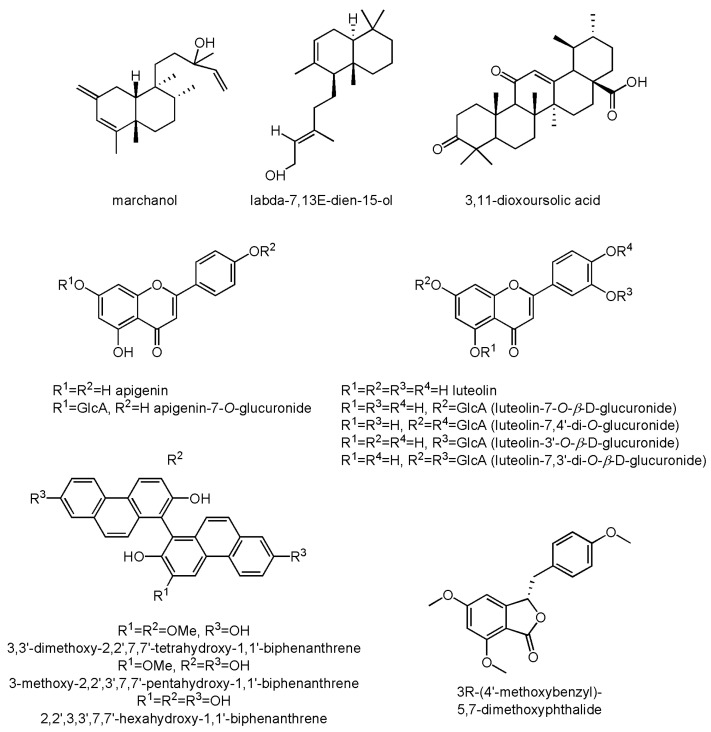
Chemical structures of some diterpenoids and triterpenoids, flavonoids, phenanthrenes, and phthalides.

**Table 1 molecules-30-00558-t001:** Secondary metabolites found in *Marchantia polymorpha*.

No.	Compounds	Formula	Geographic Origin	References
* MONOTERPENOIDS *				
1	Limonene	C_10_H_16_	USA *	[14]
* SESQUITERPENOIDS *				
**acoranes**				
2	β-Acoradiene	C_15_H_24_	Poland	[15]
3	α-Neocallitropsene	C_15_H_26_	Poland	[15]
4	β-Alaskene	C_15_H_24_	Poland	[15]
5	Acorenone B	C_15_H_24_O	Poland	[15]
**aromadendranes**				
6	α-Gurjunene	C_15_H_24_	Serbia, Turkey, USA *	[14,16,17]
7	Aromadendrene	C_15_H_24_	Turkey	[16]
8	Viridiflorol	C_15_H_26_O	Serbia	[17]
**barbatenes**				
9	α-Barbatene	C_15_H_24_	Japan	[18]
10	β-Barbatene	C_15_H_24_	Japan, Turkey	[16,18,19,20]
**bisabolanes**				
11	β-Bisabolene	C_15_H_24_	Japan	[19]
**caryophyllanes**				
12	β-Caryophyllene	C_15_H_24_	Japan	[19]
**cedranes**				
13	α-Cedrene	C_15_H_24_	Japan	[19]
14	7-epi-α-Cedrene	C_15_H_24_	Poland	[15]
15	β-Cedrene	C_15_H_24_	France	[21]
**chamigranes**				
16	α-Chamigrene	C_15_H_24_	Japan, Germany, India	[20,22,23]
17	β-Chamigrene	C_15_H_24_	Germany, India, Poland, Japan, France, Serbia, USA *	[14,15,17,19,22,23,24,25]
18	*ent*-9-oxo-α-Chamigrene (Laurencenone C)	C_15_H_22_O	Japan, Germany, Poland	[20,22,24]
**cuparanes**				
19	Cuparene	C_15_H_24_	Japan, Poland, France, Serbia, USA *	[14,15,17,18,19,25]
20	α-Cuprenene	C_15_H_24_	Japan, France, Poland	[15,19,20]
21	β-Cuprenene	C_15_H_24_	Japan, France	[19]
22	γ-Cuprenene	C_15_H_24_	Japan	[20]
23	δ-Cuprenene	C_15_H_24_	Japan, France, Poland	[15,19,20]
24	β-Microbiotene	C_15_H_24_	Poland	[15]
25	2-Cuparenol (=Cuparophenol, *δ*-Cuparenol, 2-Hydroxycuparene)	C_15_H_22_O	South Africa, Japan, France	[18,19,24,26]
26	*ent*-Cuprenenol	C_15_H_26_O	Japan, France	[19]
27	Cyclopropanecuparenol	C_15_H_26_O	Japan, France, Poland, Serbia	[15,17,19,20]
28	*epi*-Cyclopropanecuparenol	C_15_H_26_O	Japan, France, Poland	[15,19]
**cyclomyltaylanes**				
29	Cyclomyltaylenol	C_15_H_26_O	Serbia	[17]
**elemanes and bicycloelemanes**				
30	β-Elemene	C_15_H_24_	Japan	[19,21]
31	δ-Elemene	C_15_H_24_	Japan	[19,24,25]
32	Bicycloelemene	C_15_H_24_	Japan, France	[19]
**eudesmanes**				
33	α-Selinene	C_15_H_24_	Turkey, Poland, Serbia	[15,17,19]
34	*ent*-β-Selinene	C_15_H_24_	India, Japan	[19,23]
35	α-Eudesmol	C_15_H_26_O	Turkey	[16]
36	β-Eudesmol	C_15_H_26_O	Turkey	[16]
**eremophilanes**				
37	Eremophilene	C_15_H_24_	France	[21]
38	1(10),11-Eremophiladien-9β-ol	C_15_H_24_O	Germany	[27]
**germacranes**				
39	Costunolide	C_15_H_20_O_2_	Japan	[28]
**guaianes**				
40	5-Guaia-11-ol	C_15_H_26_O	Serbia	[17]
**herbertanes**				
41	β-Herbertenol	C_15_H_22_O	Japan, Poland	[15,18,19]
42	ent-α-Herbertenol	C_15_H_22_O	Germany	[22]
**himachalanes**				
43	α-Himachalene	C_15_H_24_	USA *	[14,25]
**monocyclofarnesanes**				
44	(2*Z*,4*E*)-Abscisic acid	C_15_H_20_O_3_	USA	[29]
45	(2*E*,4*E*)-Abscisic acid	C_15_H_20_O_3_	USA	[29]
**thujopsanes**				
46	*ent*-Thujopsene	C_15_H_24_	Japan, Poland, France, Serbia, USA *	[14,15,17,18,19,20,30]
47	*ent*-Thujopsan-7β-ol	C_15_H_26_O	Japan, Germany	[20,22]
48	*ent*-Thujopsenone (=Thujops-3-en-5-one)	C_15_H_22_O	Japan, France, Serbia	[17,18,19,20]
**widdranes**				
49	Widdrol	C_15_H_26_O	Japan	[18,19]
* DITERPENOIDS *				
50	Marchanol	C_20_H_32_O_2_	Vietnam	[31]
51	Labda-7,13*E*-dien-15-ol	C_20_H_34_O	Japan	[19,30,32]
52	Vitexilactone	C_22_H_34_O_4_	Vietnam	[31]
53	Phytol	C_20_H_40_O	South Africa, Poland	[15,19,33]
* STEROLS and TRITERPENOIDS *				
54	Campesterol	C_28_H_48_O	South Africa, Germany, Japan, India, Taiwan	[22,23,24,33,34]
55	Brassicasterol	C_28_H_46_O	Japan	[24]
56	Dihydrobrassicasterol	C_28_H_48_O	Taiwan	[34]
57	Stigmasterol	C_29_H_48_O	South Africa, Japan, Germany	[22,24,33]
58	Sitosterol	C_29_H_50_O	South Africa, Germany, Taiwan	[22,33,34]
59	Clionasterol (24β-ethyl)	C_29_H_50_O	Taiwan	[34]
60	12-Oleanene-3-one	C_30_H_48_O	Vietnam	[31]
61	Ursolic acid	C_30_H_48_O_3_	Vietnam	[31]
62	3,11-Dioxoursolic acid	C_30_H_44_O_4_	Vietnam	[31]
* BIBENZYLS *				
63	Lunularin	C_14_H_14_O_2_	Germany, Vietnam, China)	[19,22,31,35]
64	Lunularic acid	C_15_H_14_O_4_	Japan *, Germany	[36,37,38]
65	Prelunularic acid	C_15_H_16_O_5_	Japan	[38,39]
66	2,5-Di-*O*-β-d-glucopyranosyl- 4′-hydroxybibenzyl	C_26_H_34_O_13_	China	[40]
67	2-[3-(Hydroxymethyl) phenoxy]-3-[2-(4-hydroxyphenyl) ethyl]phenol	C_21_H_20_O_4_	China	[41]
* BISBIBENZYLS *				
68	Riccardin C (=plagiochin G)	C_28_H_24_O_4_	South Africa, India, Vietnam, China	[23,31,33,35]
69	Riccardin D (=plagiochin E)	C_28_H_24_O_4_	China	[42]
70	Riccardin G (=plagiochin E methyl ether)	C_29_H_26_O_4_	China *	[35]
71	Riccardin H	C_31_H_28_O_4_	China	[42]
72	Isoriccardin C	C_28_H_24_O_4_	India, Vietnam	[23,31]
73	Isoriccardin D	C_28_H_24_O_4_	China	[41]
74	13,13′-*O*-Isopropylidenericcardin D	C_31_H_28_O_4_	China	[42]
75	Polymorphatin A	C_28_H_24_O_4_	China	[41]
76	Marchantin A	C_28_H_24_O_5_	China, Germany, India, Japan, Serbia, Vietnam	[19,22,23,25,30,31,35,42,43,44,45,46]
77	7′,8′-Dehydromarchantin A	C_28_H_24_O_4_	Serbia *	[43]
78	Marchantin B	C_28_H_24_O_6_	China, Germany Japan	[19,22,30,35,42,43,45]
79	Marchantin C	C_28_H_24_O_4_	South Africa, Germany, India, Japan, Serbia *	[19,22,23,30,33,43,45]
80	Marchantin D	C_28_H_24_O_6_	Germany, India, China	[22,23,30,35,45,47]
81	Marchantin E	C_29_H_26_O_6_	China, Germany, India, France, Serbia *	[19,22,23,30,35,42,43,45]
82	Marchantin F	C_28_H_24_O_7_	South Africa, China	[35]
83	Marchantin G	C_28_H_22_O_6_	Japan	[47]
84	Marchantin H	C_28_H_24_O_5_	South Africa,	[33]
85	Marchantin J	C_30_H_28_O_6_	China, Germany	[22,41]
86	Marchantin K	C_29_H_26_O_7_	Germany, Vietnam, China	[22,31,35]
87	Marchantin L	C_28_H_24_O_6_	Germany	[22]
88	Isomarchantin C	C_28_H_24_O_4_	India	[23]
89	Neomarchantin A	C_28_H_24_O_4_	China	[42]
90	Perrottetin E	C_28_H_26_O_4_	China, India	[23,35,41]
* OTHER AROMATICS *				
91	3R-(3,4-Dimethoxybenzyl)- 5,7-dimethoxyphthalide	C_19_H_20_O_6_	Vietnam	[31]
92	Marchatoside	C_20_H_22_O_7_	Vietnam	[31]
93	3-(3,4-Dihydroxyphenyl)- 8-hydroxyisocoumarin	C_15_H_10_O_5_	Germany *	[37]
94	2,3-Dimethoxy-7-hydroxy- phenanthrene	C_16_H_14_O_3_	Germany *	[37]
95	2,7-Dihydroxy-3-methoxy- phenanthrene	C_15_H_12_O_3_	Germany *	[37]
96	3,3′-Dimethoxy-2,2′,7,7′-tetra- hydroxy-1,1′-biphenanthrene	C_30_H_22_O_6_	Germany *	[37]
97	2-Hydroxy-3,7-dimethoxy phenanthrene	C_16_H_14_O_3_	India	[23]
98	*m*-Hydroxybenzaldehyde	C_7_H_6_O_2_	Germany	[22]
99	*p*-Hydroxybenzaldehyde	C_7_H_6_O_2_	South Africa, Germany	[22,33]
100	3-Methoxy-2,2′,3′,7,7′-pentahydroxy- 1,1′-biphenanthrene	C_29_H_20_O_6_	Germany *	[37]
101	2,2′,3,3′,7,7′-Hexahydroxy- 1,1′-biphenanthrene	C_28_H_18_O_6_	Germany *	[37]
102	2-(3,4-Dihydroxyphenyl)-ethyl- β-d-allopyranoside	C_14_H_20_O_8_	China	[40]
103	2-(3,4-Dihydroxyphenyl)-ethyl- β-d-glucopyranoside	C_14_H_20_O_8_	China, Germany *, Japan	[37,40,48]
104	2-(3,4-Dihydroxyphenyl)-ethyl- *O*-α-l-rhamnopyranosyl-(1→2)- β-d- allopyranoside	C_20_H_30_O_12_	China	[40]
105	2-(3,4-Dihydroxyphenyl)-ethyl- *O*-β-d-xylopyranosyl-(1→6)- *O*-β-d-allopyranoside	C_19_H_28_O_12_	China	[40]
106	Salidroside	C_14_H_20_O_7_	Japan	[48]
107	Indole acetic acid	C_9_H_7_O_2_N	USA	[29]
* FLAVONOIDS *				
108	Apigenin	C_15_H_10_O_5_	Germany *, New Zealand	[37,49,50]
109	Apigenin-7-O-β-D-glucuronide	C_21_H_18_O_11_	New Zealand	[49,50]
110	Apigenin-7,4′-di-O-glucuronide	C_27_H_26_O_17_	New Zealand	[49,50]
111	Luteolin	C_15_H_10_O_6_	Germany	[22,49,50]
112	Luteolin-7-O-β-D-glucuronide	C_21_H_18_O_12_	New Zealand	[49,50]
113	Luteolin-7,3′-di-O-β-glucuronide	C_27_H_26_O_18_	New Zealand	[49,50]
114	Luteolin-7,4′-di-O-β-glucuronide	C_27_H_26_O_18_	New Zealand	[49,50]
115	Luteolin-3′4′-di-O-β-glucuronide	C_27_H_26_O_18_	New Zealand	[49,50]
116	Luteolin-3′-O-β-glucuronide	C_21_H_18_O_12_	New Zealand	[49,50]
117	Luteolin-7,3′4′-tri-O-β-glucuronide		New Zealand	[49,50]
118	Artemetin	C_20_H_20_O_8_	Vietnam	[31]
119	Kaempferol	C_15_H_10_O_6_	Vietnam	[31]
120	Quercetin	C_15_H_10_O_7_	Vietnam	[31]
121	Aureusidin-6-O-g1ucuronide	C_21_H_18_O_12_	New Zealand	[51]
122	5,3′,4′-Trihydroxyisoflavone- 7-*O*-β-d-glucopyranoside (=Orobol-7-*O*-glucoside)	C_21_H_20_O_11_	China	[40]
123	Riccionidin A	C_15_H_9_O_6_	Germany *	[52]
124	Riccionidin B	C_30_H_17_O_12_	Germany *	[52]
* LIPIDS *				
125	Palmitic acid (16:0) (=Hexadecanoic acid)	C_16_H_32_O_2_	Japan *	[20,53]
126	Ethyl palmitate (=Hexadecanoic acid ethyl ester)	C_18_H_36_O_2_	Japan	[20]
127	Stearic acid (18:0) (=Octadecanoic acid)	C_18_H_36_O_2_	Japan *	[53]
128	Palmitoleic acid (16:1n-7) (=9-Hexadecenoic acid)	C_16_H_30_O_2_	Japan *	[53]
129	Oleic acid (18:1n-9) (=9-Octadecenoic acid)	C_18_H_34_O_2_	Japan *	[53]
130	Linoleic acid (18:2n-6) (=9,12-Octadecadienoic acid)	C_18_H_32_O_2_	Japan *	[20,53]
131	α-Linolenic acid (18:3n-3) (=9,12,15-Octadecatrienoic acid)	C_18_H_30_O_2_	Japan *	[53]
132	Arachidonic acid (20:4n-6) (=5,8,11,14-Eicosatetraenoic acid)	C_20_H_32_O_2_	Japan *	[53,54]
133	EPA (20:5n-3) (=5,8,11,14,17-Eicosapentaenoic acid)	C_20_H_30_O_2_	Japan *	[53,54]
134	Oxacycloheptadecan-2-one	C_16_H_30_O_2_	Japan	[20]
* OTHER COMPOUNDS *				
135	Shikimic acid 4-(β-d-xylopyranoside)	C_12_H_18_O_9_	China	[40]

* axenic or cell culture.

**Table 2 molecules-30-00558-t002:** Distribution of sesquiterpenoids and bisbibenzyls in relation to *M. polymorpha* subspecies and geographical origin.

Characteristic Compounds	Subspecies			Geographical Origin										
	Mpr	Mpp	Mpm	Japan	Poland	Germany	France	Serbia	Turkey	India	China	Vietnam	South Africa	USA
SESQUITERPENOIDS														
Acoranes	✓				✓									
Aromadendranes		✓	✓					✓	✓					✓ *
Barbatanes				✓					✓					
Bisabolanes				✓										
Caryophyllanes				✓										
Cedranes				✓	✓		✓							
Chamigranes	✓			✓	✓	✓	✓	✓		✓				✓ *
Cuparanes	✓			✓	✓	✓	✓	✓					✓	✓ *
Cyclomyltaylanes		✓	✓					✓						
Elemanes				✓			✓							
Eudesmanes		✓	✓	✓				✓	✓	✓				
Eremophillanes		✓				✓	✓							
Germacranes				✓										
Guaianes		✓	✓					✓						
Herbertanes				✓	✓	✓								
Himachalanes														✓ *
Monocyclofarnesanes														✓
Thujopsanes	✓			✓	✓	✓	✓	✓						✓ *
Widdranes				✓										
BISBIBENZYLS														
Marchantin A	✓			✓	✓	✓		✓		✓	✓	✓		
Marchantin B				✓		✓					✓			
Marchantin C	✓			✓		✓		✓ *		✓			✓	
Marchantin D						✓				✓	✓			
Marchantin E	✓					✓	✓	✓ *		✓	✓			
Marchantin F											✓		✓	
Marchantin G				✓										
Marchantin H													✓	
Marchantin J						✓					✓			
Marchantin K						✓					✓	✓		
Marchantin L						✓								
Isomarchantin C										✓				
Neomarchantin A											✓			
Riccardin C										✓	✓	✓	✓	
Riccardin D											✓			
Riccardin G											✓ *			
Riccardin H											✓			
Isoriccardin C										✓		✓		
Isoriccardin D											✓			
Perrottetin E										✓	✓			

* cell culture; Mpr—M. polymorpha subsp. ruderalis; Mpp—M. polymorpha subsp. polymorpha; Mpm—M. polymorpha subsp. montivagans.

## Data Availability

No new data were created.

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
