# Peer review of "Marchantia polymorpha as a Source of Biologically Active Compounds"

_molecules, 2025, doi:10.3390/molecules30030558_

Round 1
Reviewer 1 Report
Comments and Suggestions for Authors
Some suggestions for revised manuscript as following:
1. Any heterogeneity within a single plant species would result in variability in chemical composition, leading to variability in biological characteristics. The foreword of the article implies that the author, along with supporting literature, should incorporate comparisons of chemical compositions between the model species and its subspecies to elucidate this aspect.
2. In the second part of the article, it is suggested that the author should present the characteristic chemical components of M. polymorpha, such as terpenoids and bisbibenzyls compounds, according to their structural types, so that readers can easily understand the structure of the compounds.
3. Within the scope of biological activity, the author predominantly explored the antibacterial effects of M. polymorpha. However, the depth of this discussion is somewhat limited. It is recommended that the author consider a more detailed examination of biological activity, specifically focusing on various compound types and their corresponding mechanisms of action. This approach would significantly enhance the readability of the text.
4. In the summarization and prospect part of the paper, the author asserts that M. polymorpha exhibits a notably diverse array of phytochemical properties, though its therapeutic potential remains largely unexplored. As such, it is recommended that the author incorporate additional references to furnish a thorough summary and perspective on the utilization of M. polymorpha chemical constituents.
5. In addition, there are many formatting errors in this article. Please refer to the published articles and revise them one by one.
Author Response
Thank you for your valuable suggestions, which helped us to improve our paper. We have now integrated the text as specified below. All adjustments are highlighted in yellow in the modified version.
Some suggestions for revised manuscript as following:
- Any heterogeneity within a single plant species would result in variability in chemical composition, leading to variability in biological characteristics. The foreword of the article implies that the author, along with supporting literature, should incorporate comparisons of chemical compositions between the model species and its subspecies to elucidate this aspect.
We added text just after Table 1. Additionally, a new Table - Table 2 - was prepared.
- In the second part of the article, it is suggested that the author should present the characteristic chemical components of polymorpha, such as terpenoids and bisbibenzyls compounds, according to their structural types, so that readers can easily understand the structure of the compounds.
We added Figures 2-5 to represent the structures of the several compound types.
- Within the scope of biological activity, the author predominantly explored the antibacterial effects of polymorpha. However, the depth of this discussion is somewhat limited. It is recommended that the author consider a more detailed examination of biological activity, specifically focusing on various compound types and their corresponding mechanisms of action. This approach would significantly enhance the readability of the text.
We added new information and modified paragraph 3. Unfortunately, data on the mechanisms of bioactivity of products of M. polymorpha are limited, particularly with reference to bacteria.
- In the summarization and prospect part of the paper, the author asserts that polymorpha exhibits a notably diverse array of phytochemical properties, though its therapeutic potential remains largely unexplored. As such, it is recommended that the author incorporate additional references to furnish a thorough summary and perspective on the utilization of M. polymorpha chemical constituents.
We added more references and a comment in the Conclusions.
- In addition, there are many formatting errors in this article. Please refer to the published articles and revise them one by one.
Format was revised conforming to the editorial style of Molecules.
Reviewer 2 Report
Comments and Suggestions for Authors
The manuscript provides insights into biologically active compounds derived from Marchantia polymorpha L. However, several points require attention to enhance the clarity, precision, and comprehensiveness of the review:
The novelty and motivation of this review must be explicitly highlighted to emphasize its contribution to the existing body of knowledge.
Table 1: Expand the table to include the biological activities of each metabolite, supplemented with appropriate latest references for validation.
Elaborate on the potential link between the geographic origin of M. polymorpha specimens and the metabolites found in the plant. Discuss how environmental factors or genetic variability might influence metabolite diversity.
The biological activities section should be significantly expanded. Include a detailed review of previous literature, comparing findings and emphasizing the significance of M. polymorpha metabolites in various biological contexts.
Additionally, a figure should be included to visually depict the various biological activities (such as antibacterial, antifungal, cytotoxic, enzyme inhibition, and insect repellence) of the metabolites.
Simplify sentences to improve clarity and eliminate redundancies. For instance:
Replace "It is worth to mention" with "It is worth mentioning."
Correct "Belonging to Marchantiaceae family" to "belonging to the Marchantiaceae family."
Revise "Proven to withhold" to "shown to possess" or "demonstrated."
Replace vague terms like "dominant compounds" and "characteristic compound" with "major secondary metabolites" and "key secondary metabolite," respectively, for better scientific precision.
Figure 1: Enhance the figure by indicating the morphological differences between female and male plants to provide a more comprehensive visual representation.
Ensure that scientific names are italicized consistently throughout the manuscript (e.g., Lines 145, 152, 279, 289).
Ensure Figures numbers.
Update the references abd focus on the latest ones.
The manuscript should be revised by a native language expert.
Author Response
Thank you for your valuable suggestions, which helped us to improve our paper. We have now integrated the text as specified below. All adjustments are highlighted in yellow in the modified version.
The manuscript provides insights into biologically active compounds derived from Marchantia polymorpha L. However, several points require attention to enhance the clarity, precision, and comprehensiveness of the review:
The novelty and motivation of this review must be explicitly highlighted to emphasize its contribution to the existing body of knowledge.
The introduction was improved. Additional information was added to this part.
Table 1: Expand the table to include the biological activities of each metabolite, supplemented with appropriate latest references for validation.
We cannot integrate information on bioactivities in Table 1 by reason of the limited space. On the other hand, the available information concerning the biological activity of individual metabolites of M. polymorpha is scarce.
Elaborate on the potential link between the geographic origin of M. polymorpha specimens and the metabolites found in the plant. Discuss how environmental factors or genetic variability might influence metabolite diversity.
We prepared an additional table – Table 2 – and integrated text in paragraph 2.
The biological activities section should be significantly expanded. Include a detailed review of previous literature, comparing findings and emphasizing the significance of M. polymorpha metabolites in various biological contexts.
We expanded section 3 by further analyzing several aspects and additional references.
Additionally, a figure should be included to visually depict the various biological activities (such as antibacterial, antifungal, cytotoxic, enzyme inhibition, and insect repellence) of the metabolites.
We cannot think of a meaningful figure to depict the bioactivities.
Simplify sentences to improve clarity and eliminate redundancies. For instance:
Replace "It is worth to mention" with "It is worth mentioning."
Correct "Belonging to Marchantiaceae family" to "belonging to the Marchantiaceae family."
Revise "Proven to withhold" to "shown to possess" or "demonstrated."
Replace vague terms like "dominant compounds" and "characteristic compound" with "major secondary metabolites" and "key secondary metabolite," respectively, for better scientific precision.
All these adjustments have been done. Moreover, the text has been revised throughout to improve the language style.
Figure 1: Enhance the figure by indicating the morphological differences between female and male plants to provide a more comprehensive visual representation.
The difference between male and female plants results from the general aspect of the plants. Such detailed morphological data are beyond the scope of this study.
Ensure that scientific names are italicized consistently throughout the manuscript (e.g., Lines 145, 152, 279, 289).
Ensure Figures numbers.
Update the references abd focus on the latest ones.
The manuscript should be revised by a native language expert.
All these checks have been done, and the reference list updated, including the most recent papers on the subject.
Reviewer 3 Report
Comments and Suggestions for Authors
The manuscript "Marchantia polymorpha as a Source of Biologically Active Compounds"provides a review of the chemical diversity and biological activities of Marchantia polymorpha. The detailed tables listing various compounds and their origins are particularly valuable for researchers in the field.
The manuscript is well-organized and covers a wide range of activities, including antifungal, antibacterial, cytotoxic, antioxidant, and anti-inflammatory properties. This breadth of coverage highlights the potential applications of M. polymorpha in various fields.
· Points for Improvement:
· I recommend the authors consider changing "biological activities" to "pharmacological activities" to better reflect the focus of their review on the therapeutic and medicinal implications of the compounds studied.
· Ensure that the species name is consistently italicized throughout the manuscript. There are instances where the species name is not italicized, such as in the abstract and introduction.
· While the manuscript provides a broad overview, some sections could benefit from a deeper analysis. For example, the discussion on the mechanisms of action for the activities could be expanded to provide more insights.
· The manuscript relies heavily on older references. Including more recent studies would strengthen the review and provide up-to-date information on the subject. For instance, PANNU, Anshul et al. Phytochemical characterization and antifungal activity of Marchantia polymorpha L. against Rhizoctonia solani. Pharmacological Research-Modern Chinese Medicine, v. 11, p. 100426, 2024.
· There are inconsistencies in the chemical nomenclature and formatting throughout the manuscript. Ensuring uniformity in the presentation of chemical names and structures would improve clarity.
· Some information is repeated in different sections, which can be streamlined. For instance, the biological activities of certain compounds are mentioned multiple times in different contexts.
· The manuscript could benefit from a more critical evaluation of the studies reviewed. Highlighting the limitations and gaps in the current research would provide a balanced perspective and suggest areas for future investigation.
· The introduction provides a good background but could be more concise. It should focus on setting the stage for the review rather than providing extensive details that are covered later.
· This section is informative but could be more engaging. Including more discussion on the significance of the chemical diversity and how it relates to the activities would add depth.
· The «Biological Activities Section’ is well-detailed but could benefit from a more structured approach. Grouping the activities by type (e.g., antimicrobial, cytotoxic) and discussing them in a systematic manner would improve readability. These effects could be summarized in a table would be a to provide a clear and concise overview of the various activities and the compounds responsible for them.
· 2. The manuscript highlights various biological activities of M. polymorpha, including antifungal, antibacterial, cytotoxic, antioxidant, and anti-inflammatory properties. These activities suggest significant therapeutic potential. However, the authors do not provide a detailed status of ongoing research or recent advancements in the therapeutic applications of M. polymorpha.
· 3. The manuscript does not mention any specific compounds or products derived from M. polymorpha that are currently available on the market for therapeutic use. This is an important aspect that could be addressed to provide a complete picture of the practical applications of the research.
· The authors discuss various in vitro studies demonstrating the biological activities of M. polymorpha compounds. However, there is no mention of any clinical or pre-clinical studies that have been conducted to evaluate the efficacy and safety of these compounds in humans or animal models. Including information on any ongoing or completed clinical trials would significantly enhance the manuscript.
· The manuscript does not explicitly address the challenges associated with developing M. polymorpha compounds into drugs. Some potential challenges that could be discussed include: Bioavailability and Pharmacokinetics, Toxicity and Safety, Toxicity and Safety. This could be included in a section of future directions for research on M. polymorpha.
· The conclusion should summarize the key findings more succinctly and provide a clear statement on the potential applications and future directions for research on M. polymorpha.
Comments on the Quality of English Language
The manuscript contains grammatical errors and awkward phrasings. A thorough proofreading and professional editing would improve readability.
Author Response
The manuscript "Marchantia polymorpha as a Source of Biologically Active Compounds" provides a review of the chemical diversity and biological activities of Marchantia polymorpha. The detailed tables listing various compounds and their origins are particularly valuable for researchers in the field.
The manuscript is well-organized and covers a wide range of activities, including antifungal, antibacterial, cytotoxic, antioxidant, and anti-inflammatory properties. This breadth of coverage highlights the potential applications of M. polymorpha in various fields.
Thank you for your positive comments and valuable suggestions, which helped us to improve our paper. We have now integrated the text as specified below. All adjustments are highlighted in the modified version.
-        Points for Improvement:
-        I recommend the authors consider changing "biological activities" to "pharmacological activities" to better reflect the focus of their review on the therapeutic and medicinal implications of the compounds studied.
We retained ‘biological activity’, since it has a broader meaning as compared to pharmacological activity. Indeed, besides pharmacological applications we also mentioned antifungal properties, not only referred to Candida albicans but towards fungal plant pathogens, too.
-        Ensure that the species name is consistently italicized throughout the manuscript. There are instances where the species name is not italicized, such as in the abstract and introduction.
This has been checked throughout the manuscript.
-        While the manuscript provides a broad overview, some sections could benefit from a deeper analysis. For example, the discussion on the mechanisms of action for the activities could be expanded to provide more insights.
We further developed section 3. However, the available data on the mechanisms of bioactivity of products of M. polymorpha are limited, particularly with reference to bacteria.
-        The manuscript relies heavily on older references. Including more recent studies would strengthen the review and provide up-to-date information on the subject. For instance, PANNU, Anshul et al. Phytochemical characterization and antifungal activity of Marchantia polymorpha L. against Rhizoctonia solani. Pharmacological Research-Modern Chinese Medicine, v. 11, p. 100426, 2024.
We updated the reference list with recently published on the subject.
-        There are inconsistencies in the chemical nomenclature and formatting throughout the manuscript. Ensuring uniformity in the presentation of chemical names and structures would improve clarity.
Chemical nomenclature has been revised in Table 1 and throughout the text.
-        Some information is repeated in different sections, which can be streamlined. For instance, the biological activities of certain compounds are mentioned multiple times in different contexts.
We revised the manuscript to remove duplication of certain information.
-        The manuscript could benefit from a more critical evaluation of the studies reviewed. Highlighting the limitations and gaps in the current research would provide a balanced perspective and suggest areas for future investigation.
We considered this suggestion in the Conclusions.
-        The introduction provides a good background but could be more concise. It should focus on setting the stage for the review rather than providing extensive details that are covered later.
The introduction was revised accordingly.
-        This section is informative but could be more engaging. Including more discussion on the significance of the chemical diversity and how it relates to the activities would add depth.
Section 2 was revised; however, to avoid duplication of information with section 3 we decided to consider the chemistry only.
-        The «Biological Activities Section’ is well-detailed but could benefit from a more structured approach. Grouping the activities by type (e.g., antimicrobial, cytotoxic) and discussing them in a systematic manner would improve readability. These effects could be summarized in a table would be a to provide a clear and concise overview of the various activities and the compounds responsible for them.
We consistently modified this section to more coherently discuss the several bioactivities and improve readability. However, we considered that a table dedicated to the bioactivities could not be necessary, as these data are only available for a few compounds.
-        2.     The manuscript highlights various biological activities of M. polymorpha, including antifungal, antibacterial, cytotoxic, antioxidant, and anti-inflammatory properties. These activities suggest significant therapeutic potential. However, the authors do not provide a detailed status of ongoing research or recent advancements in the therapeutic applications of M. polymorpha.
There are no recent advancements in this respect.
-        3.     The manuscript does not mention any specific compounds or products derived from M. polymorpha that are currently available on the market for therapeutic use. This is an important aspect that could be addressed to provide a complete picture of the practical applications of the research.
-        The authors discuss various in vitro studies demonstrating the biological activities of M. polymorpha compounds. However, there is no mention of any clinical or pre-clinical studies that have been conducted to evaluate the efficacy and safety of these compounds in humans or animal models. Including information on any ongoing or completed clinical trials would significantly enhance the manuscript.
-        The manuscript does not explicitly address the challenges associated with developing M. polymorpha compounds into drugs. Some potential challenges that could be discussed include: Bioavailability and Pharmacokinetics, Toxicity and Safety, Toxicity and Safety. This could be included in a section of future directions for research on M. polymorpha.
No products derived from M. polymorpha are currently available on the market. Also, to the best of our knowledge no clinical, pre-clinical or toxicological studies have been carried out so far. We highlighted this gap in the Conclusions.
-        The conclusion should summarize the key findings more succinctly and provide a clear statement on the potential applications and future directions for research on M. polymorpha.
This section was rewritten according to the referees’ observations and suggestions.
Round 2
Reviewer 1 Report
Comments and Suggestions for Authors
This manuscript has made many improvements according to the reviewer's comments, and I agree to accept it for publication. The Editorial Office can give more effective suggestions for improvement.
Reviewer 2 Report
Comments and Suggestions for Authors
The manuscript has been significantly improved to meet the reviewer's satisfaction. The authors have adequately addressed all the points raised.